# The In Vitro Cytotoxicity of *Eremothecium* oil and Its Components—Aromatic and Acyclic Monoterpene Alcohols

**DOI:** 10.3390/ijms23063364

**Published:** 2022-03-20

**Authors:** Elena Semenova, Victoria Presniakova, Vera Kozlovskaya, Natalia Markelova, Alexey Gusev, Wolfgang Linert, Alexander Kurakov, Anastasia Shpichka

**Affiliations:** 1Institute of Biochemical Technology, Ecology and Pharmacy, V.I. Vernadsky Crimean Federal University, 295007 Simferopol, Russia; galex0330@gmail.com; 2Institute for Regenerative Medicine, Sechenov University, 119991 Moscow, Russia; vslpp@mail.ru (V.P.); nataljamarkelova@yandex.ru (N.M.); ana-shpichka@yandex.ru (A.S.); 3Mazlumov All-Russian Research Institute of Sugar Beet and Sugar, 396030 Voronezh, Russia; kozlovskaya.vera2020@yandex.ru; 4Institute for Applied Synthetic Chemistry, Vienna University of Technology, Getreidemarkt 9/163, A-1060 Vienna, Austria; wolfgang.linert@tuwien.ac.at; 5Faculty of Biology, Lomonosov Moscow State University, 119234 Moscow, Russia; kurakov57@mail.ru; 6World-Class Research Center “Digital Biodesign and Personalized Healthcare”, Sechenov University, 119991 Moscow, Russia; 7Chemistry Department, Lomonosov Moscow State University, 119991 Moscow, Russia

**Keywords:** terpene, essential oil, *Eremothecium gossypii*, *Eremothecium ashbyi*, cytotoxicity, rose oil, 2-phenylethanol, geraniol, citral, citronellol, nerol, linalool, *Rosa damascena*, *Rosa gallica*

## Abstract

The microscopic fungi *Eremothecium ashbyi* and *E. gossypii* are known for their ability to synthetize essential oil, which has a composition similar to that of rose oil. The development of *Eremothecium* oil technology enables the production of rose-scented products, which are demanded by pharmaceutical, food, and perfumery industries. This study focuses on assessing the in vitro cytotoxicity of *Eremothecium* oil, in comparison with that of rose oil, using a combination of methods and two cell types (3T3 mouse fibroblast cell line and bone-marrow-derived mesenchymal stromal cells (BM-MSCs)). The *Eremothecium* oil samples possessed cytotoxic effects that varied among strains and batches. The revealed cytotoxicity level may be used to tailor the qualitative and quantitative composition of *Eremothecium* oil to achieve a particular quality in its end products. These results require further analysis using other cell types and assays based on measuring other cell functions.

## 1. Introduction

Rose essential oil is highly demanded and has a wide range of applications in the pharmaceutical, food, and perfumery industries. However, the quantity of this oil is limited because of a shortage of raw materials, and the rose oil industry is mainly localized in Bulgaria and Turkey. Rose oil manufacturers are trying to solve this problem by improving farming techniques, using drip irrigation systems, and adding agrochemicals to the soil. This has led to a significant rise in the flower yield [1,2,3]. Nevertheless, rose oil production remains cost-intensive, mainly because of its laboriousness and time limitations. Harvesting rose flowers is the most labor-consuming part of the process and one of the decisive factors that influences the rose oil price [3]. Thus, there has been an increasing interest in biotechnological sources of fragrant products, especially rose-scented ones [4,5].

In 1986, Bugorskiy et al. revealed that homothallic ascomycetes—*Eremothecium ashbyi* Guillermond and *E. gossypii* (Ashby et Nowell) Kurtzman (synonym for *Ashbya gossypii* (Ashby et Nowell) Guilliermond)—were able to excrete rose-scented essential oil, which had a composition similar to that of natural rose essential oil [6,7]. They showed that it was possible to produce this oil by submerging *E. ashbyi* and *E. gossypii* cultures, achieving a yield of up to 5.7 g/L. Therefore, *Eremothecium* oil can be considered as a cheap high-grade alternative that can help to reduce the industry consumption of rose oil.

To promote *Eremothecium* oil applications, it should be extensively analyzed. This includes testing its activity and safety. Previously, we showed the antimicrobial effects of essential oils synthesized by *E. ashbyi* and *E. gossypii* in comparison with rose oil and its main components [8]. We revealed that the *Eremothecium* oil’s antibiotic activity against *Escherichia coli*, *Pseudomonas aeruginosa*, *Myxococcus* sp., *Lactobacillus acidophilus*, *Lactococcus lactis* ssp. *lactis*, *Stenotrophomonas maltophilia*, *Acinetobacter baumannii*, *Klebsiella pneumoniae*, *Staphylococcus aureus*, *Bacillus subtilis*, *B. megatherium*, and *Candida albicans* was similar to that of essential oil from rose petals. These effects were ensured by both its individual components and the synergism of their combination. The analysis of the *Eremothecium* oil’s bacteriostatic activity revealed reliable positive correlations between the level of growth suppression and lag phase lengthening at a concentration range from 0.49 μL/mL to 7.81 μL/mL (R = 0.73; *p* < 0.05), as well as between the level of growth suppression of multidrug-resistant strains (*K. pneumoniae*, *P. aeruginosa*, and *A. baumannii*), and total monoterpene alcohol concentration (R = 1.0; *p* < 0.05) [9].

Moreover, there are data on the *Eremothecium* oil toxicity assessed using only *Paramecium caudatum* [10]. It was shown that the toxicity of *Eremothecium* oil at a concentration higher than 860 μg/mL reached almost 100%, while the average lethal concentration was 210 μg/mL. The severity of the effects correlated with the concentrations of phenylethanol (R = −0.9; strong negative relationship), geraniol (R = 0.6; moderate positive relationship), nerol (R = −0.55; moderate negative relationship), linalool (R = −0.74; strong negative relationship), and the total monoterpene alcohol concentration (R = 0.5; moderate positive connection) in samples [11].

Nevertheless, the lack of cytotoxicity data can be obviously observed, and more findings are urgently required, especially using human and mammalian cell cultures. Thus, this study aims to reveal the in vitro cytotoxicity of *Eremothecium* oil and its components in comparison with that of rose oil, using a combination of methods and two cell types (3T3 mouse fibroblast cell line and mesenchymal stromal cells (MSCs)).

## 2. Results

The GS analysis of the component composition of *Eremothecium* and rose oils is presented in Table 1. The tested samples of oil produced by *E. ashbyi*, except EO-03, had low variations in component composition. Compared to them, the oil produced by *E. gossypii* had a high concentration of 2-phenylethanol; however, it did not exceed that of both rose oil samples. Interestingly, EO-03 had the lowest concentration of 2-phenylethanol, which can be explained by strain-determined synthetic activity. The highest concentration of monoterpene alcohols was achieved during the culturing of *E. ashbyi* strain VKM F-3009.

The MTT and LDH assay showed that all oil samples and their individual compounds were cytotoxic and caused a dose-dependent response in both tested cell cultures (Table 2 and Table 3, Figure 1 and Figure 2). The achieved data enabled us to calculate the toxicity parameters, such as LC50 and IC50 (Table 4). Both 3T3 cells and BM-MSCs were highly susceptible to the samples tested and controls; however, BM-MSCs showed higher sensitivity than 3T3 cells. Results of the MTT assay were in high correspondence with those of the LDH assay.

The lowest cytotoxicity for both 3T3 cells and BM-MSCs was revealed in testing Crimean rose oil, which had the highest percentage of 2-phenylethanol. For all samples, the cytotoxicity of *Eremothecium* oil was revealed to be higher than that of rose oil and similar to that of individual compounds. Despite differences in component composition, all *Eremothecium* oil samples had a similar level of cytotoxicity. However, having the minimum 2-phenylethanol content (4.40%), EO-03 caused the lowest cell viability among them. The highest cell viability was revealed for cells treated with EO-06, which contained the maximum percentage of 2-phenylethanol (35.89%). This may be due to the low toxicity of 2-phenylethanol compared to other compounds (Table 2 and Table 3, Figure 2).

## 3. Discussion

There is a growing interest in using essential oils as a promising antimicrobial agent, which could possess significant effects and potentiate the action of antibiotics [12]. Their hydrophobic components can enter into the periplasm through porin proteins on the outer membrane [13,14]. By accumulating on the cell membrane, essential oils disrupt it, which leads to an increase in membrane permeability for protons and other ions, causing a shift in the intercellular pH homeostasis. Moreover, in some cases, components of essential oils can change the conformation of proteins of the membrane [15]. These effects were shown for both non-toxic (0.05 μL/mL–0.5 μL/mL) and cytotoxic (≥3.125 μL/mL) concentrations [16,17]. It should be noted that the biological activity of essential oils is ensured by a specific combination of their components, which may possess even higher efficiency than individual compounds [18]. Therefore, approaches to regulate oil biosynthesis are highly demanded. As can be observed from Table 1, *Eremothecium* oil biotechnology provides a possibility to tune the content of particular components, e.g., 2-phenylethanol and monoterpene alcohols.

Here, we tried not only to evaluate the in vitro cytotoxicity of the *Eremothecium* oil samples but also to reveal the possible impact of their individual compounds on the observed effects. We showed that the cell cultures used (3T3 cells and BM-MSCs) were highly susceptible compared to those of animals (data from safety datasheets, GOST 30333-2007, and EU 1907/2006). We used two methods to test cytotoxicity: MTT and LDH assays. The first method is based on the formation of non-water-soluble colored formazan crystals due to the NAD-dependent oxidation of tetrazolium bromide ((3-(4,5-dimethylthiazol-2-yl)-2,5-diphenyltetrazolium bromide); MTT). This method mostly reflects the cells’ metabolic activity and has several limitations. Particularly, the cell responses may overlap, and the MTT oxidation in cells may be enhanced because of the adaptation to toxic agents [19]. Moreover, the formed formazan crystals cannot be properly dissolved, leading to errors in the assessed cytotoxicity level. Therefore, to adequately estimate in vitro cytotoxicity, we also used the second method—LDH assay; the released LDH is inversely proportional to the number of damaged cells. The achieved results had high correspondence with each other. 3T3 cells are one of the standard cell lines, and BM-MSCs are a common primary cell culture that are used to assess cytotoxicity. The cell viability, after treating with samples, significantly varied compared to parameters regulated by ISO 10993-5: 2009 (except citral).

## 4. Materials and Methods

Reagents. 2-phenylethanol, linalool, nerol, citronellol, geraniol, citral (including their GC analytical standards), and solvents were purchased from Sigma-Aldrich (Taufkirchen, Germany), rose absolute sample was purchased from Neal’s Yard Remedies (Dorset, UK), and Crimean rose oil sample was purchased from Krymskaya Rosa (Simferopol, Ukraine).

Fermentation and oil extraction. *E. ashbyi* (VKM F-3009, VKM F-4565, and VKM F-4566) and *E. gossypii* (VKM F-3296) strains were stored on potato dextrose, Sabouraud’s, and Czapek’s media [20,21,22]. To prepare the inoculum, we used the glucose–peptone liquid medium (glucose—7.5 g/L; peptone—4.0 g/L; sodium succinate—2.0 g/L; potassium phosphate dibasic—0.5 g/L; inositol—0.14 g/L; pH 6.5) and cultured it for 48 h at a temperature of 20–24 °C. Then, we seeded the fermentation medium (saccharose—10.0 g/L; soybean meal—20.0 g/L; pH 7.0) using the prepared inoculum at a concentration of 1–5% and cultured for 48–72 h at a temperature of 20–24 °C. We extracted *Eremothecium* oil with hexane as a solvent, which was removed using a rotor vacuum evaporator.

Oil analysis. Samples were analyzed by gas–liquid chromatography using a chromatograph Clarus 680 (PerkinElmer, Waltham, USA) equipped with a capillary polar column (INNOWAX^®^, 60 m × 0.32 mm) according to GOST ISO 7609:2014 “Essential oils—Analysis by gas chromatography on capillary columns—General method”. The column was kept at a temperature of 80°C for 5 min, and then the temperature was gradually increased up to 250 °C at a rate of 2 °C/min. The samples (0.2 μL) were injected at a temperature of 250 °C, and compounds were detected at a temperature of 250 °C using a flame ionization detector. Helium was used as a carrier gas at a flow rate of 0.5 mL/min. The main volatile compounds (geraniol, citronellol, nerol, 2-phenylethanol, linalool, and citral) were identified using retention times of their GC analytical standards, and their mass fractions were calculated.

Cell cultures. 3T3 mouse fibroblast cell lines and bone-marrow-derived mesenchymal stromal cells (BM-MSCs) were kindly provided by the Biobank at Sechenov University (Moscow, Russia). 3T3 cells were cultured in DMEM/F12 (Invitrogen, Carlsbad, USA) medium supplemented with penicillin (100 μg/mL), streptomycin (100 μg/mL), gentamycin (50 μg/mL), and fetal bovine serum (5%, FBS, HyClone, Logan, USA). BM-MSCs were cultured in DMEM/F12 medium (Invitrogen, USA) supplemented with L-glutamine (5 mg/mL, Gibco, USA), insulin–transferrin–sodium selenite (ProSpec, Israel), basic fibroblast growth factor (bFGF) (20 ng/mL, ProSpec, Israel), and gentamycin (50μg/mL, Paneco, Russia), as well as 10% FBS, under standard conditions (37 °C; 5% CO_2_).

MTT assay. The oil samples were diluted down to concentrations of 100, 200, 400, and 800 µg/mL using dimethyl sulfoxide (DMSO) and culture medium as described elsewhere [23]. The maximum DMSO concentration did not exceed 0.5%. 3T3 cells and BM-MSCs were seeded into 96-well cell culture plates (TPP, Switzerland) at a concentration of 2 × 10^4^ cells per well and were cultured for 24 h [24]. Then, the medium was removed from the wells, and 100 μL of either the sample or sodium dodecyl sulfate (SDS) (positive control) solution was added. The cells were incubated for 24 h. Changes in cell morphology caused by the sample cytotoxicity were revealed; however, these data were not used in the quantitative analysis. The supernatant was carefully removed, and MTT (0.5 mg/mL) solution was pipetted into each well. Plates were incubated for 2 h. The formazan crystals that formed were diluted by adding 100 µL DMSO. The plates were placed onto a plate shaker and then measured using a microplate reader Titertek Multiscan (Flow Laboratories, Finland) at a wavelength of 570 nm (reference wavelength 650 nm). The cell viability was calculated as follows:(1)$$\mathrm{Viability},\;\%\;=100\times \frac{OD570e}{OD570b}\;,$$
where *OD*570*e*—optical density value when the sample was added; *OD*570*b*—optical density value of non-treated cells.

Parameters such as LC50 (median lethal concentration) and IC50 (half-maximal inhibitory concentration) were calculated according to MR 1.1.726-98.

Lactate Dehydrogenase assay. The prepared dilutions of *Eremothecium* and rose oils, as well as their components, were tested using Pierce LDH Cytotoxicity assay kit (Thermo Scientific, Waltham, USA) in accordance with a manufacturer’s instructions. 3T3 cells and BM-MSCs were seeded into plates as described above and cultured for 24 h. LDH activity was measured after adding a substrate at a wavelength of 492 nm using a microplate reader Titertek Multiscan (Flow Laboratories, Helsinki, Finland). Cells treated with 1% Triton-X100 in phosphate-buffered saline (PBS) were used as a positive control, and non-treated cells as a negative control (spontaneous LDH release). Results were calculated in percent relative to the positive control.

Statistical analysis. All experiments were performed at least three times, and the results presented are those from single experiments yielding similar results to the triplicate experiments. Each data point shows the mean ± standard deviation. The analysis was conducted using ANOVA; the differences were assumed to be statistically significant if the *p*-value was less than 0.05 [25].

## 5. Conclusions

The approach described in this study can be successfully used to achieve preliminary results and screen the activity of essential oils. The assessed in vitro cytotoxicity, using a combination of methods and two cell types, may be applied to tune the qualitative and quantitative composition of *Eremothecium* oil samples. It can also be used to improve the quality of the end products. These findings should be verified using other cell types and assays based on measuring other cell functions (e.g., PicoGreen), including cell cycle and apoptosis.

## Figures and Tables

**Figure 1 ijms-23-03364-f001:**
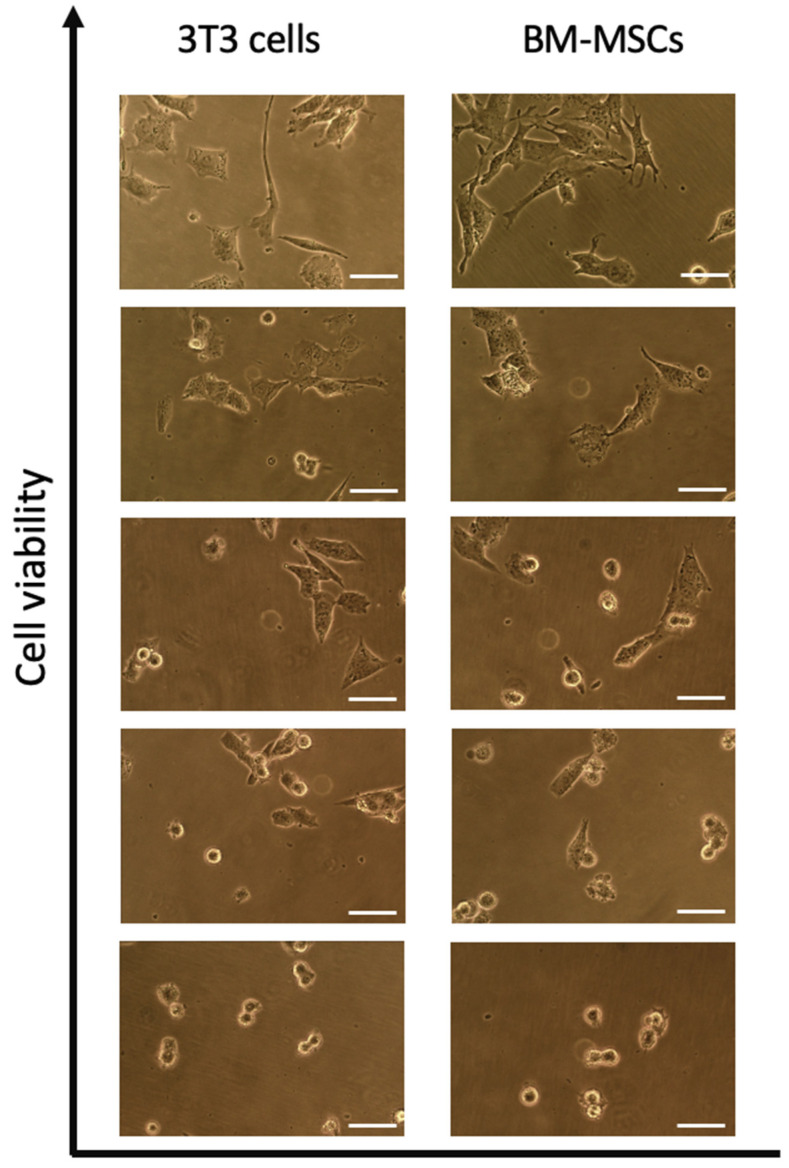
3T3 cells and BM-MSCs treated with *Eremothecium* oil samples. The top panel is non-treated cells (control). Phase-contrast microscopy. Scale bar = 30 µm.

**Figure 2 ijms-23-03364-f002:**
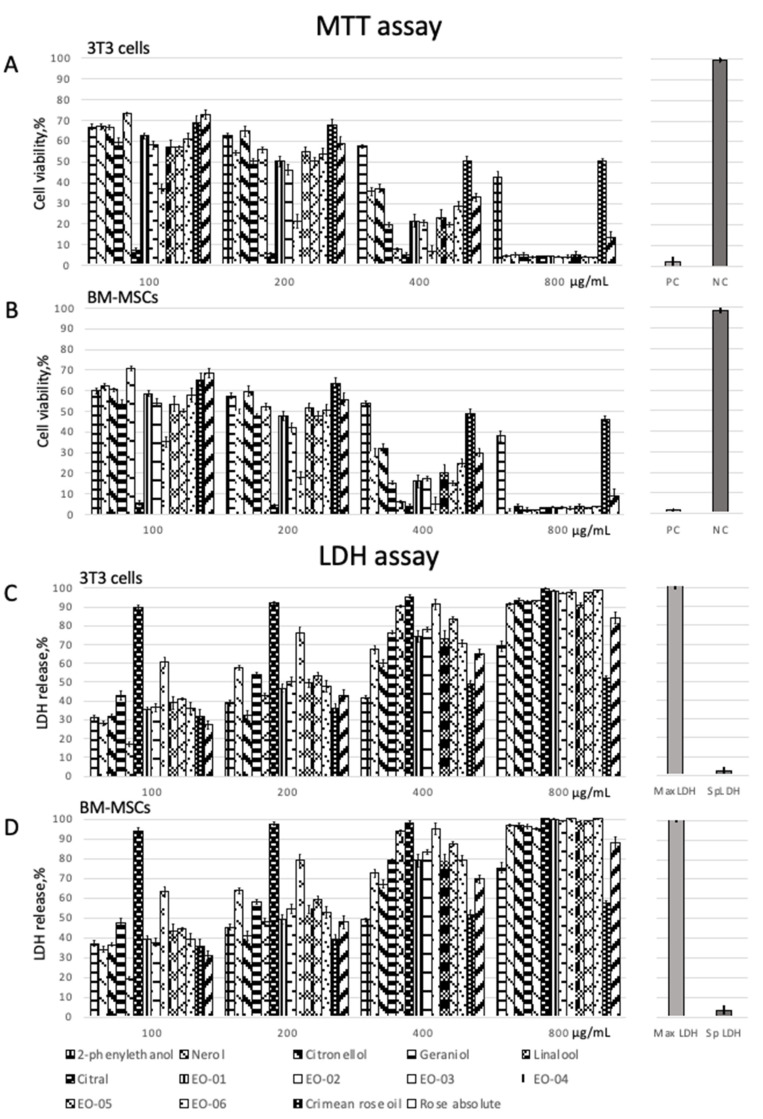
Viability of 3T3 cells (**A**) and BM-MSCs (**B**) treated with *Eremothecium* and rose oils, and their components (PC—positive control (SDS)); NC—negative control (non-treated cells). LDH release from 3T3 cells (**C**) and BM-MSCs (**D**) treated with *Eremothecium* and rose oils, and their components (MaxLDH—maximum LDH release (Triton X); SpLDH—spontaneous LDH release (non-treated cells)).

**Table 1 ijms-23-03364-t001:** Components of essential oils produced by *E. ashbyi* and *E. gossypii* (GS analysis).

Strain, Species	Sample	PEA, %	Monoterpene Alcohols, %				
			Geraniol	Citronellol	Nerol	Linalool	Total
VKM F-3009, *E.a.*	EO-01	13.23	68.84	5.66	3.99	0.34	78.83
VKM F-3009, *E.a.*	EO-02	13.85	74.66	5.64	3.60	0.25	84.15
VKM F-4565, *E.a.*	EO-03	4.40	43.30	5.50	1.85	0.01	52.15
VKM F-4566, *E.a.*	EO-04	16.32	67.17	6.22	4.02	0.83	78.24
VKM F-4566, *E.a*	EO-05	15.84	61.03	8.24	6.97	0.01	76.24
VKM F-3296, *E. g.*	EO-06	35.89	41.34	6.90	6.59	1.08	55.91
*Rosa gallica*	Crimean rose oil	71.77	15.80	3.90	5.90	0.12	25.72
*Rosa damascena*	Rose absolute	67.33	5.62	8.54	3.50	0.25	17.91

Note. *E. a.*—*E. ashbyi*; *E. g.—E. gossypii*; PEA—phenylethyl alcohol.

**Table 2 ijms-23-03364-t002:** Viability of cells treated with *Eremothecium* and rose oils and their components. MTT assay (3T3/MSCs).

	Concentration, μg/mL	Cell Viability, %			
Sample		100	200	400	800
2-phenylethanol		66.70 ± 1.38	62.50 ± 1.50	57.45 ± 0.96	42.69 ± 2.62
		60.06 ± 2.08	57.43 ± 1.87	53.87 ± 1.34	38.02 ± 1.98
Nerol		67.03 ± 1.38	54.40 ± 1.29	35.72 ± 2.13	4.27 ± 0.56
		62.34 ± 1.74	49.98 ± 2.32	30.16 ± 1.87	2.43 ± 0.69
Citronellol		66.88 ± 1.08	64.77 ± 2.47	37.26 ± 2.02	5.21 ± 1.18
		60.45 ± 1.78	59.79 ± 2.03	32.07 ± 2.51	4.01 ± 1.24
Geraniol		59.38 ± 2.37	50.28 ± 1.28	19.74 ± 0.94	5.03 ± 1.16
		53.46 ± 2.12	47.85 ± 1.09	15.42 ± 1.83	2.10 ± 0.96
Linalool		73.20 ± 1.00	55.77 ± 1.54	8.00 ± 0.64	4.01 ± 0.45
		70.76 ± 1.79	52.34 ± 2.04	6.08 ± 0.91	1.98 ± 0.89
Citral		7.38 ± 1.33	5.49 ± 0.74	5.13 ± 1.00	4.29 ± 0.29
		5.67 ± 1.23	4.49 ± 0.98	4.12 ± 1.64	3.23 ± 0.54
EO-01		62.54 ± 1.50	50.21 ± 2.40	21.51 ± 3.10	4.38 ± 0.34
		58.64 ± 1.87	47.65 ± 2.09	16.27 ± 2.17	3.47 ± 0.39
EO-02		58.07 ± 1.99	46.19 ± 2.35	20.61 ± 1.24	4.11 ± 0.40
		54.06 ± 1.65	42.19 ± 2.04	17.56 ± 1.71	3.33 ± 0.69
EO-03		37.05 ± 2.45	21.31 ± 3.31	6.57 ± 2.90	4.21 ± 0.70
		35.18 ± 1.69	17.76 ± 1.87	5.32 ± 1.24	3.09 ± 0.76
EO-04		57.21 ± 3.50	55.02 ± 2.10	23.09 ± 4.08	5.38 ± 1.18
		53.61 ± 2.74	51.89 ± 1.86	19.96 ± 2.78	4.03 ± 1.56
EO-05		56.98 ± 0.80	50.20 ± 2.10	19.45 ± 1.10	4.16 ± 0.35
		50,18 ± 1,21	47,71 ± 1,32	15,40 ± 1,75	3,18 ± 0,59
EO-06		60.90 ± 3.20	53.90 ± 2.80	28.89 ± 2.10	4.01 ± 0.18
		57.83 ± 2.03	50.67 ± 1.94	24.91 ± 1.83	3.87 ± 0.67
Crimean rose oil		68.65 ± 3.40	68.02 ± 2.50	50.58 ± 2.10	50.43 ± 1.30
		65.21 ± 2.43	63.74 ± 2.03	48.83 ± 1.99	46.21 ± 3.52
Rose absolute		72.93 ± 2.10	59.10 ± 3.09	33.01 ± 1.96	13.23 ± 3.20
		68.54 ± 1.81	55.92 ± 2.47	29.97 ± 1.58	9.15 ± 2.32

**Table 3 ijms-23-03364-t003:** LDH release in cells treated with *Eremothecium* and rose oils, and their components (3T3/MSCs).

	Concentration, μg/mL	LDH Release, %			
Sample		100	200	400	800
2-phenylethanol		31.02 ± 0.87	39.15 ± 2.71	41.79 ± 1.54	69.18 ± 2.03
		37.12 ± 0.92	45.18 ± 1.23	49.02 ± 1.01	75.32 ± 2.94
Nerol		28.17 ± 0.96	57.93 ± 1.53	67.32 ± 1.98	91.54 ± 1.12
		34.01 ± 1.27	63.72 ± 0.65	72.65 ± 1.12	96.89 ± 2.01
Citronellol		31.96 ± 1.47	32.65 ± 2.39	60.19 ± 3.10	93.46 ± 1.88
		36.52 ± 1.91	41.07 ± 1.54	67.12 ± 2.47	97.19 ± 3.20
Geraniol		43.10 ± 2.15	54.21 ± 1.34	76.34 ± 1.12	92.57 ± 1.66
		47.34 ± 2.89	58.13 ± 1.78	79.21 ± 1.97	96.22 ± 1.09
Linalool		16.99 ± 1.19	42.85 ± 1.78	90.59 ± 1.54	93.18 ± 2.70
		19.33 ± 1.56	48.17 ± 1.87	93.71 ± 2.03	95.05 ± 2.54
Citral		89.76 ± 1.94	92.21 ± 1.62	95.40 ± 1.31	99.81 ± 0.79
		94.28 ± 2.01	97.70 ± 0.79	98.13 ± 1.34	102.34 ± 2.02
EO-01		35.53 ± 1.33	46.87 ± 1.71	74.29 ± 2.49	98.63 ± 0.67
		39.18 ± 1.77	49.19 ± 1.29	79.05 ± 1.23	99.98 ± 0.96
EO-02		36.54 ± 1.43	50.19 ± 2.01	78.32 ± 1.56	97.34 ± 0.96
		37.63 ± 1.08	54.76 ± 0.96	83.33 ± 1.41	99.01 ± 1.01
EO-03		60.92 ± 1.87	76.18 ± 2.33	91.29 ± 1.17	98.03 ± 0.64
		63.10 ± 1.55	79.09 ± 2.03	95.14 ± 1.76	100.21 ± 0.93
EO-04		38.96 ± 1.90	49.51 ± 1.67	73.38 ± 3.21	91.12 ± 1.94
		43.21 ± 1.04	53.99 ± 1.88	78.13 ± 2.05	98.36 ± 1.45
EO-05		41.08 ± 1.08	53.39 ± 1.13	83.45 ± 1.65	97.59 ± 1.83
		44.50 ± 0.96	59.04 ± 1.59	87.62 ± 1.23	99.19 ± 2.32
EO-06		36.00 ± 2.06	47.82 ± 1.19	70.61 ± 1,54	99.07 ± 1.19
		39.02 ± 1.28	52.90 ± 3.03	79.27 ± 0.94	100.31 ± 1.18
Crimean rose oil		31.95 ± 1.59	36.07 ± 1.83	48.99 ± 2.90	52.32 ± 1.69
		35.68 ± 1.32	39.06 ± 0,99	51.82 ± 2,16	57.23 ± 1.33
Rose absolute		27.39 ± 1.78	43.19 ± 2.19	65.34 ± 1.67	84.19 ± 2.01
		31.03 ± 1.37	47.82 ± 1,93	69.71 ± 1.01	87.90 ± 1.75

**Table 4 ijms-23-03364-t004:** In vitro toxicity parameters of *Eremothecium* and rose oils and their components (aromatic and monoterpene alcohols) (vs. in vivo *^1–4^).

Sample	LC50, μg/mL	LC50 *^3^, μg/mL	IC50, μg/mL	EC50 *^4^, μg/mL	LD50 *, μg/mL	
					Orally ^1^	Dermally ^2^
2-phenylethanol	568.74	<464.00	150	287.0–490.0	1600	>2000
Nerol	243.31	20.3	200	32.4	4500	>5000
Citronellol	299.91	14.66	225	17.48	3450	2650
Geraniol	157.54	3.45–22.00	150	7.75–13.1	3600	>5000
Linalool	216.18	27.80	150	59.0–88.3	2790	5610
Citral	-	6.78	75	6.8–103.8	3450–6800	2000–2250
EO-01	139.21	-	125	-	-	-
EO-02	185.27	-	150	-	-	-
EO-03	12.02	-	5	-	-	-
EO-04	182,63	-	165	-	-	-
EO-05	144.78	-	125	-	-	-
EO-06	212.71	-	150	-	-	-
Crimean rose oil	707.82	-	300	-	<12600	3000
Rose absolute	303.33	-	150	-	>5000	2500

Note. *—according to the safety datasheets, GOST 30333-2007, and EU 1907/2006. LC50—median lethal concentration; IC50—half-maximal inhibitory concentration; EC50—half-maximal effective concentration; LD50—median lethal dose. 1—rat; 2—rabbit; 3—fish; 4—daphnia.

## Data Availability

All data generated are included in the manuscript.

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
