# Peer review of "The In Vitro Cytotoxicity of Eremothecium oil and Its Components—Aromatic and Acyclic Monoterpene Alcohols"

_ijms, 2022, doi:10.3390/ijms23063364_

Round 1

Reviewer 1 Report

Review comments

The study by Elena Semenova et al on “The cytotoxicity of eremothеcium oil and its components: aromatic and acyclic monotherpene alcohols” is an interesting subject. However, some of my major concerns are

  1. English language has to be drastically improved by letting a native English language editor make corrections in the sentences wherever grammatically wrong.
  2. what are the benefits of extracting this oil from the fungi ashbyi and E. gossypii. Explain this in detail in the introduction. Describe more about the uses of eremothеcium oil in industry, healthcare, food industry and so forth.
  3. In the abstract the authors mentioned about increase in the cell metabolic activity under stress conditions of exposure at minimum toxic concentrations. Please provide the full necessary data to validate your points.
  4. For evaluating the cytotoxicity of different drugs or in your case the eremothеcium oil, how were the oils dissolved in the media. There is a very high possibility the oils may not dissolve in the media and thereby the cytotoxic effects to the 3T3 cells may be drastically reduced because of dissolving issue. Please explain in detail.
  5. For evaluating the cytotoxic effects of the compounds many times MTT assay does not accurately determine the cell death. Many times the formed formazon crystals are not dissolved properly and hence the final absorbance value is greatly altered and the final results can be flawed. I request the authors to take a look at other alternatives as well to accurately evaluate the cytotoxicity assay study. Please refer to this section and make amends

https://www.promega.kr/en/resources/pubhub/is-your-mtt-assay-really-the-best-choice/

  1. I also recommend that the authors to perform other assays such as lactate dehydrogenase assay. And provide the morphological images of the cells after treatment with eremothеcium oil at the desired time point that was used.
  2. In table 1, the authors have shown the different component composition of essential oil produced by ashbyi and E. gossypii strains. Which experiment was used for isolation of the different components and which experiment was used for identification of these different components.
  3. I recommend the authors to evaluate the cytotoxicity on human fibrobalst cell lines and better show the cytotoxicity of these oils on numerous cell lines along with the morphological images.
  4. Please explain in detail how was the LD50 determined orally and dermally in experimental animals. Why is there no mention of which experimental animal was used. Was it mice? Which strain of mice was used and how long did the mice survived after administration of the different components of the eremothеcium oil until 50% of the mice died (Table 3)
  5. The result section need to be increased and more data has to be provided and please provide the necessary information and data of the experiment animals used. Also evaluate the important biochemical markers for toxicity using the experimental mice blood or serum.
  6. The manuscript needs major improvement before being considered for revision.

Author Response

Dear Editor,

Thank you for allowing us the opportunity to improve and resubmit our manuscript “The in vitro cytotoxicity of eremothеcium oil and its components – aromatic and acyclic monoterpene alcohols”.

Please find enclosed the revised manuscript for further consideration. The manuscript has been revised according to the comments raised by the reviewers to the best of our ability. Changes to the manuscript are highlighted within the document by using the red color.

Please find a detailed reply to the reviewers’ comments attached with this revision. We would like to thank the reviewers for the constructive and competent criticism, and we hope that you will find the paper suitable for publication in “International Journal of Molecular Sciences”.

Sincerely yours,

Dr Anastasia Shpichka

REFEREE 1

The study by Elena Semenova et al on “The cytotoxicity of eremothеcium oil and its components: aromatic and acyclic monotherpene alcohols” is an interesting subject. However, some of my major concerns are

  1. English language has to be drastically improved by letting a native English language editor make corrections in the sentences wherever grammatically wrong.

Answer: The manuscript was extensively revised, and most of the text was re-written.

  1. what are the benefits of extracting this oil from the fungi ashbyi and E. gossypii. Explain this in detail in the introduction. Describe more about the uses of eremothеcium oil in industry, healthcare, food industry and so forth.

Answer: The requested information was added.

  1. In the abstract the authors mentioned about increase in the cell metabolic activity under stress conditions of exposure at minimum toxic concentrations. Please provide the full necessary data to validate your points.

Answer: The abstract was re-written.

  1. For evaluating the cytotoxicity of different drugs or in your case the eremothеcium oil, how were the oils dissolved in the media. There is a very high possibility the oils may not dissolve in the media and thereby the cytotoxic effects to the 3T3 cells may be drastically reduced because of dissolving issue. Please explain in detail.

Answer: The information was added. To increase oil solubility in the medium, we used the procedure described in (https://doi.org/10.1016/j.indcrop.2014.04.009).

  1. For evaluating the cytotoxic effects of the compounds many times MTT assay does not accurately determine the cell death. Many times the formed formazon crystals are not dissolved properly and hence the final absorbance value is greatly altered and the final results can be flawed. I request the authors to take a look at other alternatives as well to accurately evaluate the cytotoxicity assay study. Please refer to this section and make amends (https://www.promega.kr/en/resources/pubhub/is-your-mtt-assay-really-the-best-choice/)

Answer: We performed additional experiments using LDH-assay. The issue described was mentioned in the Discussion.

  1. I also recommend that the authors to perform other assays such as lactate dehydrogenase assay. And provide the morphological images of the cells after treatment with eremothеcium oil at the desired time point that was used.

Answer: The data requested were added.

  1. In table 1, the authors have shown the different component composition of essential oil produced by ashbyi and E. gossypii strains. Which experiment was used for isolation of the different components and which experiment was used for identification of these different components.

Answer: The requested information was added.

  1. I recommend the authors to evaluate the cytotoxicity on human fibrobalst cell lines and better show the cytotoxicity of these oils on numerous cell lines along with the morphological images.

Answer: We performed additional experiments using bone-marrow derived mesenchymal stromal cells. The data were added to the manuscript.

  1. Please explain in detail how was the LD50 determined orally and dermally in experimental animals. Why is there no mention of which experimental animal was used. Was it mice? Which strain of mice was used and how long did the mice survived after administration of the different components of the eremothеcium oil until 50% of the mice died (Table 3)

Answer: We did not perform experiments with animals. The data provided were taken from safety datasheets, GOST 30333-2007, and EU 1907/2006.

  1. The result section need to be increased and more data has to be provided and please provide the necessary information and data of the experiment animals used. Also evaluate the important biochemical markers for toxicity using the experimental mice blood or serum.

Answer: The section was significantly expanded.

  1. The manuscript needs major improvement before being considered for revision.

Answer: The manuscript was extensively revised, and most of the text was re-written.

Reviewer 2 Report

There is a spelling mistake in the title ‘therpene’

I miss the data of positive controls used in this work. Please add these data.

I wonder about the sample “49 h” (in Figure 3, Table 2). It is unclear to me what does this mean and where does this sample come from. Please clarify this.

Lines 42/43 The sentence

“This required a comprehensive research aromatic products different quantitative composition.” needs some improvements

Line 86: in vivo should be in italics (check the whole manuscript)

Lines 89/192/251: in vitro should be written in italics (check the whole manuscript)

Lines 103/104: General method: not General method”

Line 135: gentamycin

Figure 2: geraniol and not heraniol

Line 240: formazan

Line 241: check the name 3- (4,5-dimethyl-240 tyazol-2- Il) -2,5-diphenylterazolia bromide

How were the compounds in the eremothecium oil identified? I only can find the statement about the analysis

Table 1: Is there any explanation about the low conc of PEA in sample BKMF-4565, E.a. 49h?

Author Response

Dear Editor,

Thank you for allowing us the opportunity to improve and resubmit our manuscript “The in vitro cytotoxicity of eremothеcium oil and its components – aromatic and acyclic monoterpene alcohols”.

Please find enclosed the revised manuscript for further consideration. The manuscript has been revised according to the comments raised by the reviewers to the best of our ability. Changes to the manuscript are highlighted within the document by using the red color.

Please find a detailed reply to the reviewers’ comments attached with this revision. We would like to thank the reviewers for the constructive and competent criticism, and we hope that you will find the paper suitable for publication in “International Journal of Molecular Sciences”.

Sincerely yours,

Dr Anastasia Shpichka

REFEREE 2

There is a spelling mistake in the title ‘therpene’

Answer: the spelling was corrected.

I miss the data of positive controls used in this work. Please add these data.

Answer: the data were added.

I wonder about the sample “49 h” (in Figure 3, Table 2). It is unclear to me what does this mean and where does this sample come from. Please clarify this.

Answer: We are sorry for this mistype. The required corrections were done.

Lines 42/43 The sentence “This required a comprehensive research aromatic products different quantitative composition.” needs some improvements

Answer: The sentence was removed.

Line 86: in vivo should be in italics (check the whole manuscript)

Answer: Corrected.

Lines 89/192/251: in vitro should be written in italics (check the whole manuscript)

Answer: Corrected.

Lines 103/104: General method: not General method”

Answer: We have corrected and used the official name of the standard.

Line 135: gentamycin

Answer: Corrected.

Figure 2: geraniol and not heraniol

Answer: Corrected.

Line 240: formazan

Answer: Corrected.

Line 241: check the name 3- (4,5-dimethyl-240 tyazol-2- Il) -2,5-diphenylterazolia bromide

Answer: Corrected.

How were the compounds in the eremothecium oil identified? I only can find the statement about the analysis

Answer: The information was added.

Table 1: Is there any explanation about the low conc of PEA in sample BKMF-4565, E.a. 49h?

Answer: The explanation was added.

Round 2

Reviewer 1 Report

The authors have clearly answered all my queries. However, I request the authors to add Line numbers next time wherever the manuscript has been revised and highlighted for easy evaluation by reviewers during revision. Accept it in present form.

Author Response

Dear Editor,

Thank you for allowing us the opportunity to improve and resubmit our manuscript “The in vitro cytotoxicity of eremothеcium oil and its components – aromatic and acyclic monoterpene alcohols”.

Please, find enclosed the revised manuscript for further consideration. The manuscript has been revised according to the comments raised by the reviewers to the best of our ability. Changes to the manuscript are highlighted within the document by using the red color.

Please, find a detailed reply to the reviewers’ comments attached with this revision. We would like to thank the reviewers for the constructive and competent criticism, and we hope that you will find the paper suitable for publication in “International Journal of Molecular Sciences”.

Sincerely yours,

Dr Anastasia Shpichka

REFEREE 1

The authors have clearly answered all my queries. However, I request the authors to add Line numbers next time wherever the manuscript has been revised and highlighted for easy evaluation by reviewers during revision. Accept it in present form.

ANSWER: The authors would like to sincerely thank the reviewer for his/her valuable comments that helped to significantly improve the paper!

Reviewer 2 Report

in vitro should be written in italics

Author Response

Dear Editor,

Thank you for allowing us the opportunity to improve and resubmit our manuscript “The in vitro cytotoxicity of eremothеcium oil and its components – aromatic and acyclic monoterpene alcohols”.

Please find enclosed the revised manuscript for further consideration. The manuscript has been revised according to the comments raised by the reviewers to the best of our ability. Changes to the manuscript are highlighted within the document by using the red color.

Please find a detailed reply to the reviewers’ comments attached with this revision. We would like to thank the reviewers for the constructive and competent criticism, and we hope that you will find the paper suitable for publication in “International Journal of Molecular Sciences”.

Sincerely yours,

Dr Anastasia Shpichka

REFEREE 2

in vitro should be written in italics

ANSWER: This issue was corrected in the revised version of the manuscript (in red). The authors would like to sincerely thank the reviewer for his/her valuable comments that helped to significantly improve the paper!
